

# Reconstruction of spatially detailed global map of NH$_4^+$ and NO$_3^-$ application in synthetic nitrogen fertilizer

Kazuya Nishina[*1], Akihiko Ito[1], Naota Hanasaki[1], and Seiji Hayashi[1]

[1]National Institute for Environmental Studies, Tsukuba, Ibaraki 305-8604, Japan

*Correspondence to:* Kazuya Nishina (nishina.kazuya@nies.go.jp)

**Abstract.**

This paper provides a method for constructing a new historical global nitrogen fertilizer application map ($0.5° \times 0.5°$ resolution) for the period 1961–2010 based on country-specific information from Food and Agriculture Organization statistics (FAOSTAT) and various global datasets. This new map incorporates the fraction of NH$_4^+$ (and NO$_3^-$) in N fertilizer inputs by utilizing fertilizer species information in FAOSTAT, in which species can be categorized as NH$_4^+$ and/or NO$_3^-$-forming N fertilizers. During data processing, we applied a statistical data imputation method for the missing data (19% of national N fertilizer consumption) in FAOSTAT. The multiple imputation method enabled us to fill gaps in the time-series data using plausible values using covariates information (year, population, GDP, and crop area). After the imputation, we downscaled the national consumption data to a gridded cropland map. Also, we applied the multiple imputation method to the available chemical fertilizer species consumption, allowing for the estimation of the NH$_4^+$/NO$_3^-$ ratio in national fertilizer consumption. In this study, the synthetic N fertilizer inputs in 2000 showed a general consistency with the existing N fertilizer map (Potter et al. 2010) in relation to the ranges of N fertilizer inputs. Globally, the estimated N fertilizer inputs based on the sum of filled data increased from 15 Tg-N to 110 Tg-N during 1961–2010. On the other hand, the global NO$_3^-$ input started to decline after the late 1980s and the fraction of NO$_3^-$ in global N fertilizer decreased consistently from 35% to 13% over a 50-year period. NH$_4^+$ based fertilizers are dominant in most countries; however, the NH$_4^+$/NO$_3^-$ ratio in N fertilizer inputs shows clear differences temporally and geographically. This new map can be utilized as an input data to global model studies and bring new insights for the assessment of historical terrestrial N cycling changes. Datasets available at doi:10.1594/PANGAEA.861203.

## 1 Introduction

Terrestrial nitrogen cycling is significantly governed by human activities (Galloway et al., 2004). The estimated global N loadings to ecosystems likely overload their N capacity (Rockström et al., 2009; Richardson et al., 2009; De Vries et al., 2013; Steffen et al., 2015). The anthropogenic reactive



N (Nr) loadings to terrestrial ecosystems have become twice that of the estimated biogenic N fixation in terrestrial ecosystems in the late 20th century (Gruber and Galloway, 2008). Synthetic nitrogen fertilizers and fossil fuel combustion are, respectively, the first and second largest anthropogenic sources of Nr to terrestrial ecosystems (Galloway et al., 2003, 2008). Excessive Nr in the applies

synthetic N fertilizer remains in the environment without crop uptake (Liu et al., 2010; Conant et al., 2013) and contributes to an overabundance of Nr – eutrophication, N saturation – in various ecosystems (Galloway et al., 2003). Also, anthropogenic Nr alters atmosphere–biosphere Nr exchanges via compounds such as $N_2O$, $NH_3$, and NOx (Mosier et al., 1998; Galloway et al., 2004; Davidson, 2009; Tian et al., 2016).

The use of synthetic nitrogen fertilizers grew rapidly after the birth of the Haber–Bosch technique and especially in the latter half of the 20th century (Galloway et al., 2003; Erisman et al., 2008; Sutton and Bleeker, 2013). The N fertilizer applied to cropland soils –Nr– transfers and is lost to the environment (atmosphere, hydrosphere) with an environment-specific retention time (Galloway et al., 2003). The fertilizer Nr input to terrestrial ecosystems is expected to increase during the

21st century due to increases in the world population and economic growth (Erisman et al., 2008; Davidson, 2012; Bouwman et al., 2013). However, there are still large uncertainties with regards to the historical cumulative Nr impacts on terrestrial ecosystem at the global scale. The N loading capacity of terrestrial ecosystems differs among regions due to differing climates and capacities of ecosystems (Steffen and Smith, 2013; De Vries et al., 2013). Therefore, spatiotemporal information

may be important for assessing Nr impacts on terrestrial N cycling. Currently, global synthetic N fertilizer maps are available (Potter et al., 2010; Mueller et al., 2012) only for the year 2000.

The type of synthetic fertilizer is an important factor affecting gaseous Nr release and leaching from croplands because different types of synthetic fertilizers have different chemical characteristics and bioavailability to microbes and crops. For example, when considering the amount of

$NH_3$ volatilization from N fertilized soils, whether the fertilizer contains ammonium salts ($NH_4^+$) or whether it is a $NH_4^+$-forming N fertilizer such as urea is a critical regulation factor in relation to Nr loss (Harrison and Webb, 2001; Bouwman et al., 2002). Also, the type of fertilizer (e.g., nitrate salts, ammonium salts and urea, or anhydrous $NH_4^+$) determines other N oxide gas emissions from fertilized soil (Bouwman, 1996; Harrison and Webb, 2001; Shcherbak et al., 2014). Early work by

Matthews (1994) considered the types of N fertilizer in FAO statistics for the global estimation of $NH_4$; however, to the authors' knowledge, there are no available historical N fertilizer maps that consider the type of N on a global basis. To assess Nr impacts on terrestrial N cycling in more detail, historical maps of the amounts of $NH_4^+$ and $NO_3^-$, as well as N deposition for different N forms, are required (Shindell et al., 2013).

Global national statistics (national survey data) consist of time-series cross-section datasets (i.e., those with years of data for each country). The major issue in the available national statistics for N fertilizer (FAOSTAT, IFA dataset) is that the reported data has many missing values for a significant





number of country-years. Excluding those countries with insufficient observations would reduce the number of available countries and lead to a fragmented global historical map. Regular gridded spatiotemporally continuous data of $NH_4^+$ and $NO_3^-$ are essential for global terrestrial models including biogeochemical models, crop growth models, and many others. To avoid the missing data problem, various data imputation methods have recently been developed. In particular, for time-series cross-section datasets, a statistical imputation method called "Amelia" (King et al., 2001) has succeeded in effectively filling missing values in social studies (e.g. Ross, 2006; Evans et al., 2010) and natural sciences (Pyšek et al., 2015). This method would also be suitable for missing values in an N fertilizer consumption dataset.

In this study, to facilitate historical N impact studies at a global scale, we produced a new global N fertilizer input map. This was a spatiotemporal explicit map for 1960–2010 that considered the fraction of $NH_4^+$ and $NO_3^-$ in the N fertilizer inputs. Using country statistics in FAOSTAT (Food and Agriculture Organization of the United Nations, 2014) for 1960–2010 (subsequently filled using the statistical imputation method), we downscaled the data into $0.5° \times 0.5°$ maps with historical cropland maps (Hurtt et al., 2011). To evaluate such maps, we compared them with the existing global N fertilizer map for the year 2000 and an N deposition map (Shindell et al., 2013).

## 2 Materials and methods

We used country-based statistics of N fertilizer consumption from the FAOSTAT data (Food and Agriculture Organization of the United Nations, 2014). We summarize the data processing protocol for the total N fertilizer map in Fig 1. The detailed protocol is as follows.

### 2.1 Manual cleaning and processing of FAOSTAT data

All available data for each country during the years 1961–2010 were used in this study. Issues in the reported FAOSTAT data (e.g., data referring to imports only or data expressed in formulated products) were evaluated case by case and unreliable data were considered as missing data. For some countries (e.g., the former Soviet Union, Socialist Federal Republic of Yugoslavia, Eritrea, Ethiopia, and Czechoslovak Republic), the national statistics were separated due to the change in the framework of these nations from 1960 to 2010. In order to maintain consistency in the national statistics before and after the change in political system, we used the framework in 2000. To avoid discontinuity in the statistics before and after the frame change, we used the trend in the farmer framework statistics and then weighted the N fertilizer consumptions.

### 2.2 Data imputation for missing data in the national fertilizer consumption statistics

Before the downscaling processes for the global N fertilizer map, we applied statistical data imputation to the FAOSTAT data. This was done because there were many missing values, especially for



developing countries. For data imputation, we used the multiple data imputation method for time-series cross-section datasets proposed by King et al. (2001). This was based on a bootstrapped-based expectation-maximization algorithm with an assumption of a multivariate normal distribution among covariates. The basic concept of statistical data imputation is as follows

$$D \sim \mathcal{N}_k(\mu, \Sigma) \tag{1}$$

where $D$ is a dataset with $n$ observations and $k$ variables. $\mathcal{N}_k(\mu, \Sigma)$ indicates $k$ dimensions of a multivariate normal distribution with a mean vector $\mu$ and $k \times k$ covariance matrix $\Sigma$.

In statistical data imputation, the missing dataset is assumed to be the conditional probability of the observed dataset ($D^{obs}$), which is equivalent to the conditional probability of the whole dataset $D$ (including the missing dataset $D^{miss}$) as follows

$$p(M|D) = p(M|D^{obs}) \tag{2}$$

where $M$ indicates the $n \times k$ missingness matrix, that is, each element equals 1 if the corresponding element of $D$ is missing and equals 0 if it is observed. This is based on the assumption that data are missing at random. According to this assumption, the estimation of parameters in the multivariate normal distribution for $D^{obs}$ enables the estimation of $D^{miss}$. In this study, we used the R package "Amelia" for statistical imputation (Honaker et al., 2011). This program has an advanced statistical imputation protocol to handle time-series and cross-sectional features of panel data; this is preferable to nation-based statistics. Amelia incorporates time-series smoothing for the imputed data using various algorithms such as polynomial, spline, and locally weighted scatterplot smoothing functions. In this package, to estimate $\mu, \Sigma$ for $D^{obs}$, a bootstrap-based EM algorithm was used by optimizing $L(\mu_{ob}, \Sigma_{ob}|D^{obs})$ as a posterior with flat priors (Honaker and King, 2010). Therefore, the iterative sampling from the posterior of the parameter enables the generation of multiple datasets for the missing data. Across the multiple datasets, the variance in the imputed values by bootstrapping reflects Amelia's uncertainty over the observation's true value.

We applied this algorithm to the national N fertilizer consumption dataset in FAOSTAT. The covariates used were national GDP, national population, and national cropland area. The historical cropland area in each was calculated from the Harmonized Global Land Use map (LUHa) v1.0 (Hurtt et al., 2011). Before application of the data imputation, we converted the N fertilizer consumption to a proportion (0 to 1) by dividing the maximum N fertilizer value during 1960–2010 in each country. This procedure enabled us to constrain the imputed values in the N consumption to less than the existing values. Therefore, we used a logistic transformation for the imputation in Amelia to satisfy the multivariate normal distribution. Furthermore, we used a three-order polynomial function to smooth the missing values over time within a country (Honaker and King, 2010). In this study, we



generated 1000 "complete" datasets and then took an ensemble average and standard deviation for
the generated dataset as an imputation dataset.

### 2.3 Downscaling the filled dataset of the national inventory to a spatially explicit map

To downscale the imputed national N fertilizer consumption data to the $0.5° \times 0.5°$ grid-based map,
we used the cropland fraction of each gridcell in the LUHa v1.0 map, which provides land-use maps
for 1901–2007 (Hurtt et al., 2011). For the period 2008–2010, we repeatedly used the 2007 map in
LUHa v1.0. Before the assignment of N fertilizer inputs in each grid, we calculated the time-series
of N fertilizer rates for each country $k$. This can be calculated from the balance between national N
fertilizer consumption and national cropland area in each year (see example in Fig 2). In this study,
we considered double cropping regions at both rates and timings of N fertilizer input as follows:

$$\mathrm{Ndose}_{k,t} = \frac{\mathrm{Ncons}_{k,t}}{\sum\limits_{i,j} \mathrm{Carea}_{k,t,i,j} W_{i,j}}, \quad k \in i,j \tag{3}$$

where $\mathrm{Ndose}_{k,t}$ indicates the N fertilizer input rate (kg-N ha$^{-1}$) for country $k$ at year $t$. $t$ (ranges
from 1961 to 2010). $Ncons_{k,t}$ is the national N fertilizer consumption (kg-N) of country $k$ at year
$t$. Imputed data was used for $\mathrm{Ncons}_{k,t}$ in place of missing data. $\mathrm{Carea}_{k,t,i,j}$ is the cropland area for
country $k$ at year $t$, which was based on the LUHa v1.0 map. $W_{i,j}$ indicates the weight matrix at
latitudinal i and longitudinal j positions in the corresponding country $k$. If the position at $i,j$ is a
double cropping region, the value is 2; otherwise, the value is 1 for $W_{i,j}$. This is based on the crop
calendar for double cropping species in the SAGE dataset (Sacks et al., 2010), which considered
winter (spring) barely, winter (spring) oats, winter (spring) wheat, second rice crop, second maize
sorghum, and second sorghum in this study (see map in protocol 3 of Fig 1). In this region, we
doubled the weighting of the crop area using $W_{ij}$ during each entire period.

### 2.4 Assignment of fertilizer input date using crop calendar map

Using M3-Crops Data (Monfreda et al., 2008), we determined the dominant cropping species in
each grid cell from a choice of 19 species (barely, cassava, groundnuts, maize, millet oats, potato,
pulsenes, rapeseed, rice, rye, sorghum, soybean, sunflower, sweetpotato, wheat, and yam). Next, we
used the Crop Calendar in the SAGE dataset (Sacks et al., 2010) and determined the schedule of N
fertilizer input in each grid cell using the dominant crop species. The selected species is the most
common species between the two databases. For the double cropping regions, we considered seven
cropping species, as described in the previous section (2.3). The maps for the sowing/transplanting
date are shown in Fig. 1 in protocol 4. To merge the crop calendar maps and fertilizer input map,
we applied linear interpolation to the main crop calendar map (except for cases of double cropping
regions) against each latitudinal band. This procedure enabled us to avoid a mismatch of cropland



area between the maps. For the assignment of fertilizer inputs, base fertilizer application was set at 7 days before sowing/transplanting and second fertilizer application was set at 30 days after the base fertilizer application. We assumed the ratio between the base and second fertilizer applications was 7:3. In the grid cell for the double cropping regions, we assigned "Ndose$_{k,t}$" twice for each cropping duration. The dates of N fertilizer inputs were fixed during 1960–2010.

## 2.5 Allocation of total N fertilizer dose to $NH_4^+$ and $NO_3^-$ inputs

The statistics for various types of synthetic fertilizer consumption are available in FAOSTAT. These can be sorted by the content (forming) of $NH_4^+$ and $NO_3^-$. We converted total N fertilizer input to $NH_4^+$ and $NO_3^-$ inputs based on the fertilizer species composition. Table 1 summarizes the N-based ratio of $NH_4^+$ and $NO_3^-$ contents (or forming in soils) in each fertilizer species. Before the imputation, we manually applied data cleaning, i.e., unreliable data (e.g., one-digit inflation for just one year) were discarded and treated as missing data. If information about a certain fertilizer type in a certain country was unavailable, we replaced zero instead of missing value during such periods.

In the same manner as for the national fertilizer consumption, we applied Amelia to the missing data for the multiple fertilizer species consumption. However, for the fertilizer species, we did not apply the data conversion to the proportion. Instead, we used square root transformation in the Amelia program to maintain positive values in the imputed values. For the covariates, we used the imputed national N fertilizer consumption (obtained as above) and 10 types of fertilizer species as in Table 1. Also, we used a third-order polynomial function to smooth the missing values over time within a country using Amelia (Honaker and King, 2010). We generated 1000 "complete" datasets and then took an ensemble average and standard deviation of the generated dataset as the imputation dataset.

After imputation, we calculated the ratio of $NH_4^+$ to $NO_3^-$ from the sum of each fertilizer species according to contents of $NH_4^+$ and $NO_3^-$ (Table 1) as follows:

$$\text{NH4frac}_{k,t} = \frac{\sum_m a_m \text{Fert}_{k,t,m}}{\sum_m \text{Fert}_{k,t,m}}, \quad k \in i,j \tag{4}$$

where $m$ indicates the type of fertilizer considered in this study (Table 1), NH4frac is the $NH_4^+$ fraction for each $k$ country at year $t$, $a_m$ is the coefficient of $NH_4^+$ fraction to total N in each $m$ fertilizer species (see in Table 1), and $\text{Fert}_{k,t,m}$ indicates the amount of consumption of each imputed fertilizer species for each country in year $t$. When information about a country was not available, we used the smoothed regional average, for which the regions were defined as Asia, Australia, Europe, North America, Latin America, North Africa, Sub-Sahara, and Pacific Islands.

Finally, we applied spline-smoothing to $\text{NH4frac}_{k,t}$ in the filled time-series dataset in each country $k$ to ensure smoothness over time. This is because even the existing data for each fertilizer species



seemed unreliable (e.g., round figures, 1 or 2 significant figures) and $\mathrm{NH4frac}_{k,t}$ was peaky across
years in most countries. This procedure obtained smoothness over time for each country and also
preserved trends in the consumption of fertilizer species.

## 3   Results and discussion

The key characteristics of the developed N fertilizer map are twofold: (1) it includes time-series
information owing to statistical imputation of missing records in the fertilizer census, and (2) it
includes the $\mathrm{NH_4^+}/\mathrm{NO_3^-}$ ratio. In the following sections, we introduce and validate the developed N
fertilizer map, focusing particularly on these two characteristics.

### 3.1   Visual checking of imputed data and their trends

Examples of the imputed data are visualized in Fig 3. The percentage of missing values in the na-
tional N consumption data was 16% of a total of 9811 values. The countries reported with missing
data accounted for a small fraction of N fertilizer consumption compared with global total fertilizer
use. As a result of statistical data imputation, the mean of the imputed data was seemingly reason-
able for the major countries (Fig 3). The missing data in FAOSTAT actually did not follow a random
pattern, e.g., all data for Azerbaijan before 1990 was missing. However, even for such countries, the
mean imputed data given by Amelia seemed to makes sense. For example, the trend in the mean
imputation data in Azerbaijan was similar to that in Turkmenistan (Fig 3). Amelia allows smooth
time trends, shifts across cross-sectional units, and correlations over time and space in the impu-
tation (Honaker and King, 2010). Furthermore, Amelia can take advantage of relevant information
(population, GDP, and cropland area), which retains smoothness in the time-series information over
time. In this way, the use of multiple imputation meant that we were able to generate consecutive
time-series fertilizer inputs for all countries existing between 1961 and 2010.

### 3.2   Comparison with existing N fertilizer map

We compared a new map for the year 2000 with the existing global N fertilizer map provided by
Potter et al. (2010) (hereafter "Potter's map") in Figs. 4 and 5. On the basis of the global scale N
fertilizer input, the range of fertilizer input rates in our map (99.9th quantile; 172 kg-N $\mathrm{ha}^{-1}$ $\mathrm{yr}^{-1}$)
generally agreed with the range in Potter's fertilizer map (99.9th quantile; 138 kg-N $\mathrm{ha}^{-1}$ $\mathrm{yr}^{-1}$)
and the values were well correlated with each other (R = 0.80) with no large biases (Fig 4). For the
global total N input, our study's value in 2000 was 85 Tg-N $\mathrm{yr}^{-1}$, which was larger than the global
total input of Potter's fertilizer map (70 Tg-N $\mathrm{yr}^{-1}$). Without the imputed values, the global total N
input was 81 Tg-N in 2000 in FAOSTAT. The major difference between our map and Potter's map
is in the handling of cropping species. A bottom-up approach was taken to produce Potter's map;
i.e., crop-specific fertilizer rates were first addressed and the map was produced as a summation



of all crops using each crop cultivation area map. In our study, we determined the values based only on national consumption data in FAOSTAT. In addition, the statistics in our map are based on FAOSTAT, whereas Potter et al. (2010) mainly used the International Fertilizer Industry Association

(IFA) database (http://www.fertilizer.org/statistics) as a primary information source. An additional difference is that we applied statistical data imputation to the missing data. These discrepancies yielded the 11 Tg-N difference in total N fertilizer consumption in the global outcome.

With attention to spatial differences (Fig 4), there were clear positive and negative differences in specific countries that have large cropland areas (i.e., United States, China, and India). In our map,

N fertilizer input in India shows generally higher values than those in Potter's map. In China and USA, positive and negative differences were observed regionally. Potter's fertilizer map was based on the bottom-up approach with weighted crop-specific N fertilizer inputs; therefore, N fertilizer inputs are reflected in the spatial cropping distributions of each cropping species of the M3-crops database (Monfreda et al., 2008). Also, there are differences in the cropland area maps; we used

Hurtt et al. (2011) in this study, whereas Potter et al. (2010) used Ramankutty and Foley (1999). This also resulted in spatial contrasts between the two maps. In addition, there are differences in the treatment of double cropping regions. Potter's map did not consider the double cropping regions in the estimation of fertilizer inputs. These differences should be considered as uncertainty sources when the map is used.

### 3.3 Spatial and temporal patterns of $NH_4^+$:$NO_3^-$ ratio in fertilizer map

The $NH_4^+$/$NO_3^-$ ratio in national fertilizer varied considerably both geographically and temporally (Figs. 6–9). From the $NH_4^+$/$NO_3^-$ ratios among the regions (Fig 6) and individual countries (Fig 7), certain distinct features are noticeable. For example, Chile shows an almost 100% $NO_3^-$ fraction in the 1960s (Fig 7). This is because Chilean nitrate made from desert minerals was the main source

of N fertilizer product in this period. This is a particularly striking example; however, we can see the relatively high $NO_3^-$ fraction all over Latin America during this period (Fig 6). Such types of N fertilizer were gradually replaced by urea; thus explaining the increase in the $NH_4^+$ fraction in this period. On the other hand, North America (United States and Canada) shows a lower $NH_4^+$ fraction during the entire period. In Asia, China and India show high $NH_4^+$ fractions during 1961–2010.

China mainly consumed ammonium-forming fertilizers such as urea and ammonium bicarbonate ("Other N" in FAOSTAT) (Zhang et al., 2011). Likewise, India mainly consumed urea, ammonium phosphate, and ammonium sulfate nitrate; these are $NH_4^+$-forming N fertilizers.

The global total N fertilizer input based on the sum of the dataset filled by imputation shows rapid increases during 1961-2010, reaching 110 Tg-N in 2010 (Fig 10). The historical changes in $NH_4^+$

and $NO_3^-$ inputs show different trends from the global total input. The estimated global total $NO_3^-$ input peaked in 1989 (19.7 Tg-N year$^{-1}$) and decreased to 9.9 Tg-N year$^{-1}$ in 2010. From 1961 to 2010, the fraction of $NO_3^-$ to total N fertilizer input consistently decreased from 35% to 13%



globally. On the other hand, the total amount and fraction of $NH_4^+$ increased consistently during the same period (Fig 10). This is because of the expansion of the countries with high $NH_4^+$ consumption during this period, rather than the changes in $NH_4^+$:$NO_3^-$ ratio in each country (Fig 6).

### 3.4 Comparison of N deposition with $NH_4^+$ and $NO_3^-$ inputs

Finally, we compared differences in the historical geographic distribution between our map and terrestrial N deposition (Shindell et al., 2013) for terrestrial $NH_4^+$ and $NO_3^-$ inputs. During the study period, the global annual N deposition increased from 41.9 Tg-N to 67.1 Tg-N (Fig. 11) and the global annual N fertilizer input increased from 11.7 Tg-N to 110.0 Tg-N (Fig. 10). The order of total inputs between the two Nr sources was comparable, but the rate of increase of N fertilizer input (2.43 Tg-N year$^{-1}$) was higher than that of N deposition (0.63 Tg-N year$^{-1}$). During the period, the global annual N deposition as $NO_3^-$ inputs ranged from 19.4 Tg-N to 31.1 Tg-N and annual global $NO_3^-$ inputs by fertilizer ranged from 4.1 Tg-N to 19.7 Tg-N (Fig 10). Therefore, global total $NO_3^-$ inputs from N fertilizer were consistently lower than those from N deposition.

In most latitudinal zones, total N fertilizer inputs increased due to the increase of N fertilizer consumption accompanied by crop area expansion during 1960–2010 (Fig. 11). A similar trend is seen in $NH_4^+$ fertilizer input. On the other hand, the amount of latitudinal $NO_3^-$ input in N fertilizer in 2001 became smaller compared to that in 1991. In particular, in 45°N to 55°N, $NO_3^-$ input decreased from 6.4 Tg-N to 4.4 Tg-N and further decreased to 3.2 Tg-N by 2010 (data not shown). However, this is still of a comparable order to $NO_3^-$ deposition from latitudinal inputs in the temperate to cool temperate regions of the northern hemisphere.

### 4 Conclusion and remarks

Overall, using the statistical imputation, we were able to generate a filled dataset for national fertilizer consumption for all the countries in the FAOSTAT database between 1961 and 2010. This procedure also enabled us to estimate the fraction of $NH_4^+$ ($NO_3^-$) in the total N fertilizer inputs based on chemical fertilizer information.

The products proposed in this paper could be widely utilized for global N cycling studies such as N gas emission (e.g., Hudman et al., 2012) and N leaching (e.g., He et al., 2011) on a global scale. We have a few options for the historical global N fertilizer map at present. Hence, this map could be an alternative option for estimating historical N fertilizer inputs in global model studies, although the N dose in our map is based on a simple balance equation using national consumption data and cropland area.

The ratio of $NH_4^+$ to $NO_3^-$ in N fertilizer has not been considered in earlier biogeochemical models for global N cycling. The ratio of $NO_3^-$ to total N fertilizer inputs decreased with time from 1961 to 2010; however, the amount of $NO_3^-$ as a fertilizer input is still comparable with terrestrial $NO_3^-$



deposition. In fertilized soils, $NH_4^+$ and $NO_3^-$ affect biogeochemical processes (e.g., nitrification and denitrification). Thus, in relation to $NO_3^-$, our map offers new insights into terrestrial N inputs and impacts from synthetic fertilizer.

The national census has inherent uncertainty, even in the existing activity data (Winiwarter and Muik, 2010). Therefore, to some extent, this N fertilizer map also has unavoidable uncertainties. For example, using this map as a prior distribution, inverse modeling studies with atmospheric N concentrations(e.g. Thompson et al., 2014a, b) enable us to improve and update the accuracy.

The map of the ratio of $NH_4^+$ to $NO_3^-$ in N fertilizer is also provided and this information could 305 be used in the existing N fertilizer maps. All products ($NH_4^+$ input, $NO_3^-$ input, fraction of $NH_4^+$ to total inputs, and dates for fertilizer input) are provided in the NetCDF format at https://doi.pangaea. de/10.1594/PANGAEA.861203.

*Acknowledgements.* With great respect, KN appreciate all data providers for their effort in developing and sharing valuable datasets. Our work is based entirely on these resources; we hope to have added value to these 310 existing datasets, especially to the biogeochemical modeling communities. This work is partially supported by the Environment Research and Technology Development Fund (S-10) of the Ministry of the Environment, Japan.



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



**Table 1.** Composition of N types in each fertilizer specie that appeared in FAOSTAT

| Fertilizer type | $NH_4^+:NO_3^-$ | Global cons$^*$ [Tg] |
|---|---|---|
| Ammonium nitrate | 50:50 | 5.46 |
| Ammonium phosphate | 100:0 | 4.64 |
| Ammonium sulphate | 100:0 | 2.67 |
| Ammonium sulphate nitrate | 75:25 | 0.02 |
| Calcium cyamide | 100:0 | 0.02 |
| Calcium ammonium nitrate | 50:50 | 3.25 |
| Calcium nitrate | 0:100 | 0.10 |
| Sodium nitrate | 0:100 | 0.02 |
| Urea | 100:0 | 40.20 |
| Other N | 100:0 | 15.67 |

Other N (e.g., N, P, K mixture; Nitrogenous fertilizer). The ratio of Other N is an
assumed value. $^*$Global consumption is the values in year 2000 in FAOSTAT without
the application of statistical imputation.



1. Manual data cleaning and quality check for FAOSTAT

2. Data imputation for missing data for N consumption in FAOSTAT

| Year | Country | GDP | Population | Crop land Area | Total N consumption |
|------|---------|--------|------------|----------------|---------------------|
| 1960 | A | 200000 | 9860 | 75028 | 1200 |
| 1961 | A | 202000 | 10056 | 75333 | 300 |
| … | A | 203000 | 10261 | 75637 | NA |
| … | A | 205000 | 10475 | 75942 | 299 |
| … | A | … | … | … | NA |
| 1960 | B | … | 20961 | 192236 | NA |
| 1961 | B | … | 21297 | 199541 | NA |
| … | B | … | 21633 | 206845 | 100 |
| … | B | … | … | … | 200 |

Replaced by imputed data

3. Downscaling the imputed fertilizer country statistics to grid based crop area map (weighted by double cropping regions).

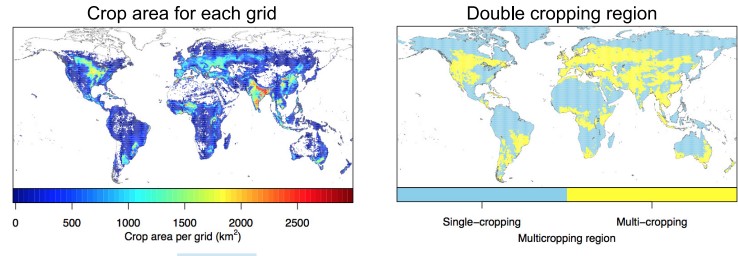

Crop area for each grid          Double cropping region

4. Using crop calendar map, assign the date of fertilization before the sowing or transplanting to each grid by each dominant crop species. In this step, the date of additional fertilizer is set to 45 days after the first (base) fertilizer.

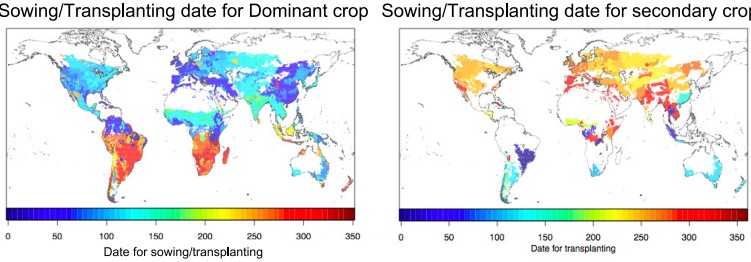

Sowing/Transplanting date for Dominant crop   Sowing/Transplanting date for secondary crop

**Figure 1.** Flow diagrams for downscaling national N fertilizer consumption data to gridded map in this study.





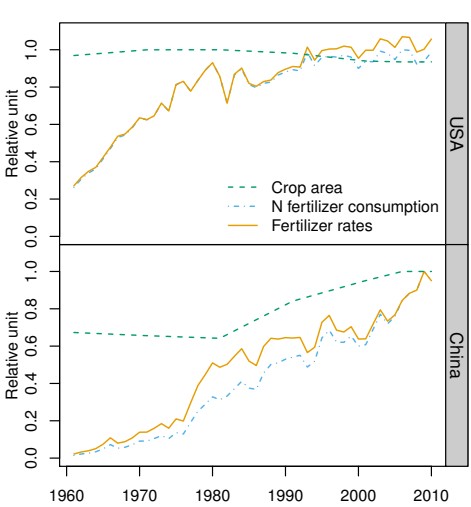

**Figure 2.** Example of determination of time-series fertilizer application rates in USA and China. For both national crop area and N fertilizer consumption, the values are divided by the maximum for this period.




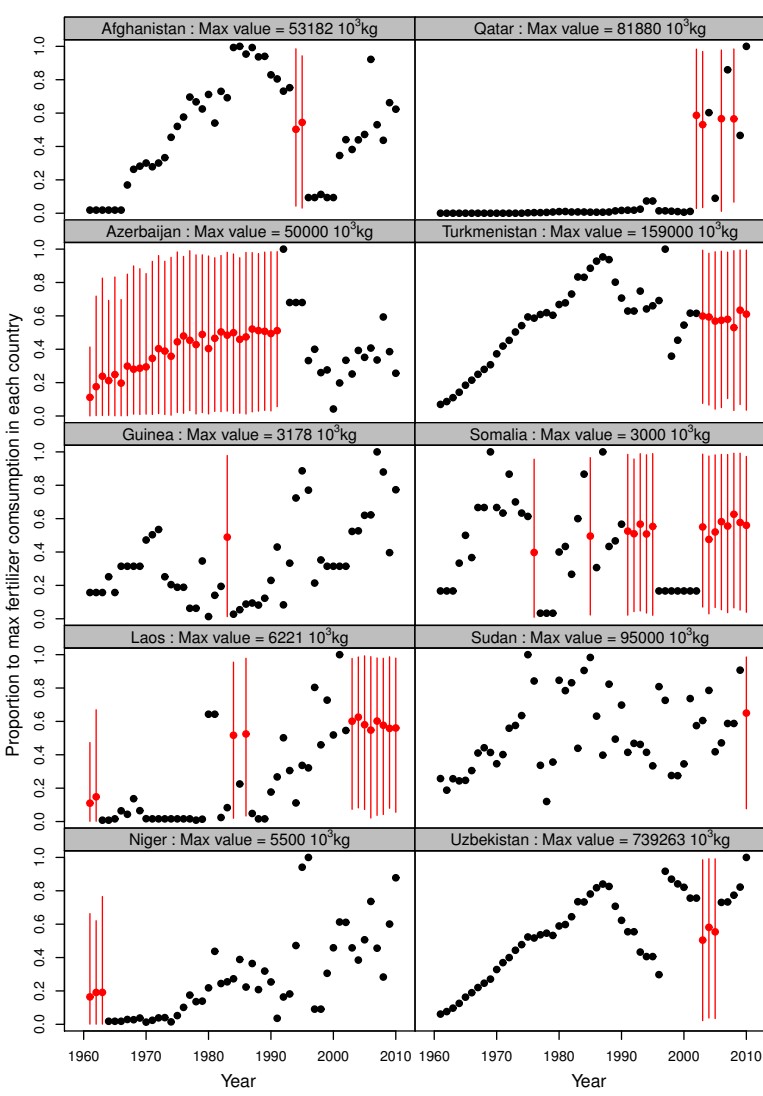

**Figure 3.** Example of imputation for missing data in national fertilizer consumption data in FAOSTAT. Black points are observed values of the time series and red points are the mean imputed data for missing values. Error bars indicate the 95% confidence interval obtained using 1000-times bootstrapped data.





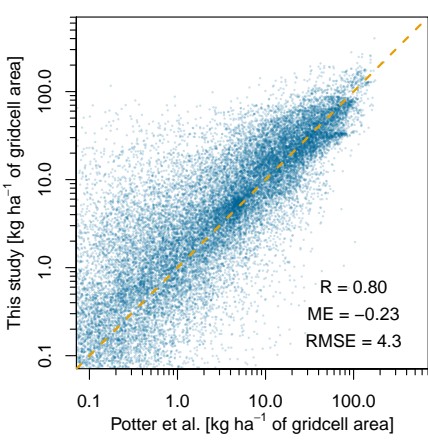

**Figure 4.** Comparison of N fertilizer input in year 2000 with data of Potter et al. (2010). Dashed orange line indicates the 1:1 line. The unit of ME and RMSE is kg-N ha$^{-1}$.



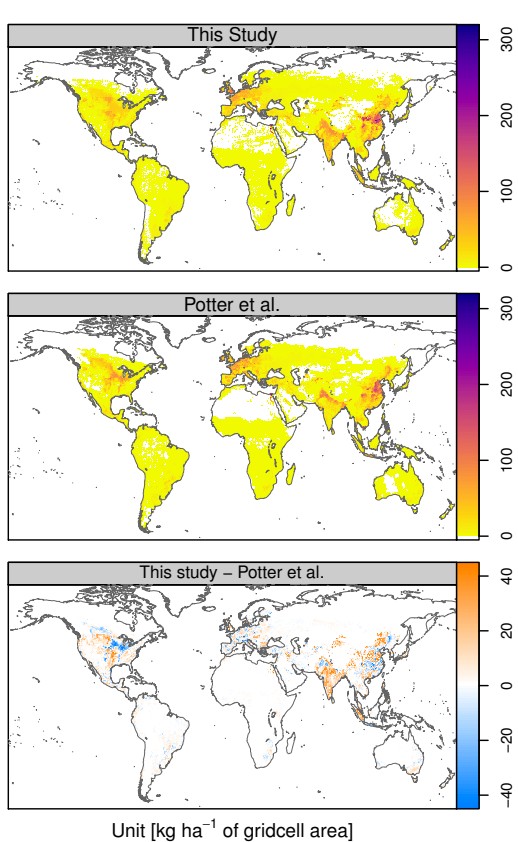

**Figure 5.** Comparison of N fertilizer inputs in year 2000





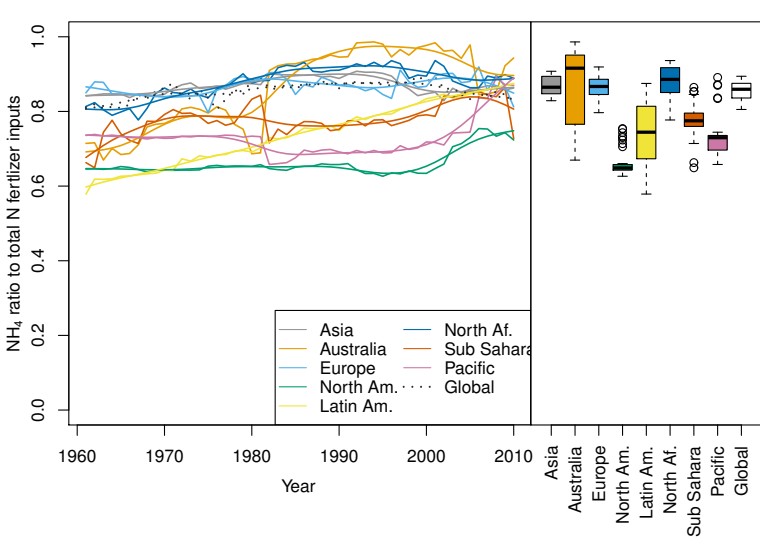

**Figure 6.** Regional fraction of $NH_4^+$ in N fertilizer input during 1961–2010. The boxplot summarizes the values over the entire period. Open circles represent outliers if the largest (or smallest) value is greater (or less) than 1.5 times the box length from the 75 % percentile (or 25 % percentile).





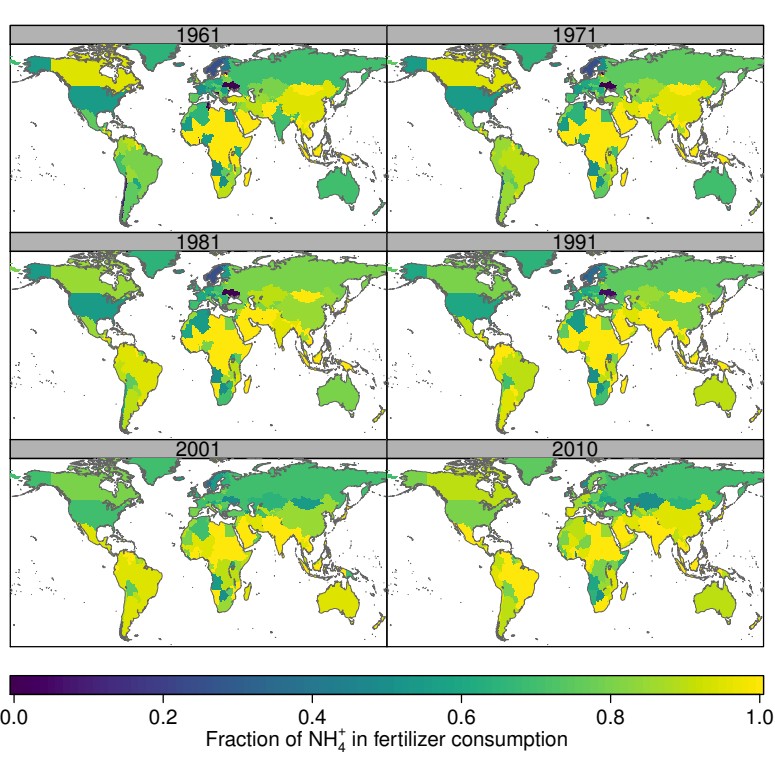

**Figure 7.** Fraction of $NH_4^+$ in N fertilizer input map during 1961–2010





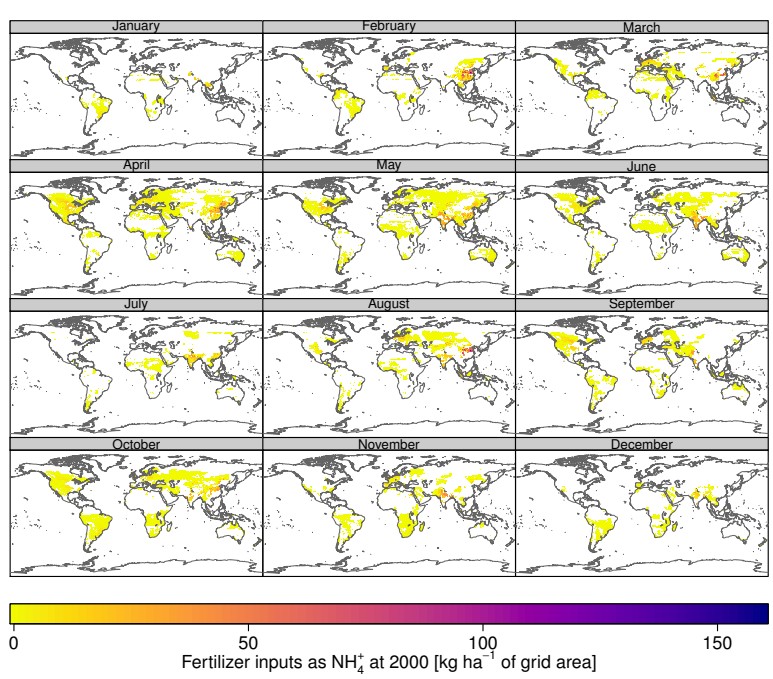

**Figure 8.** Monthly N fertilizer input as $NH_4^+$ in year 2000. Values represent average N applied over all crops across each $0.5° \times 0.5°$ grid cell





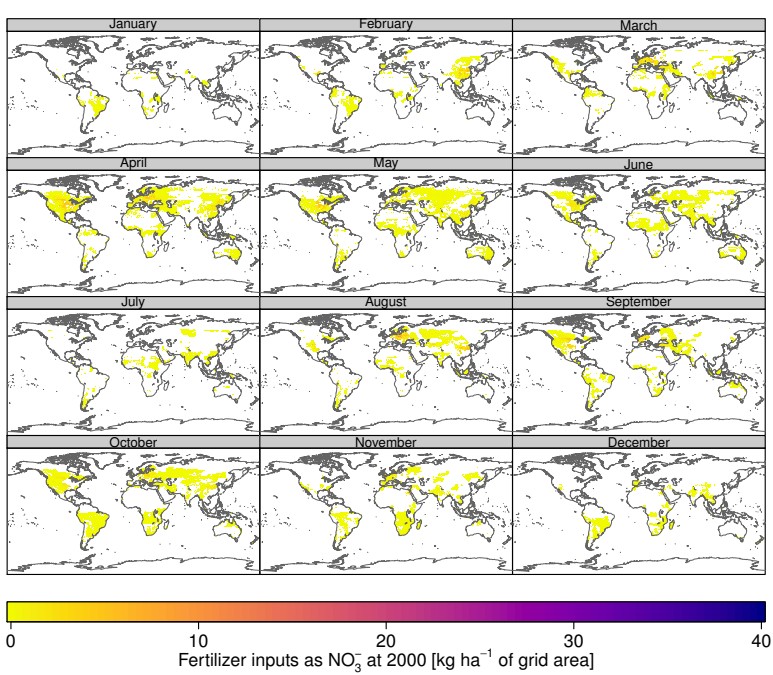

**Figure 9.** Monthly N fertilizer input as $NO_3^-$ in year 2000. Values represent average N applied over all crops across each $0.5° \times 0.5°$ grid cell.





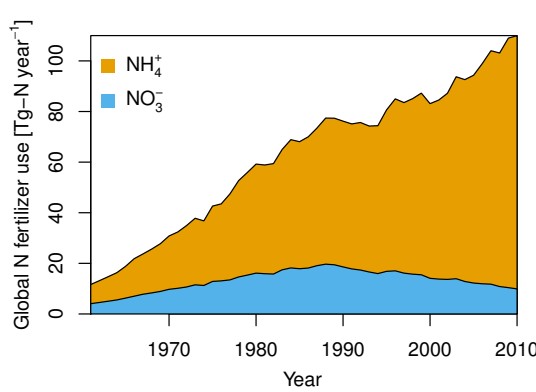

**Figure 10.** Estimated global N fertilizer use as $NH_4^+$ and $NO_3^-$ inputs based on the sum of the imputed dataset.



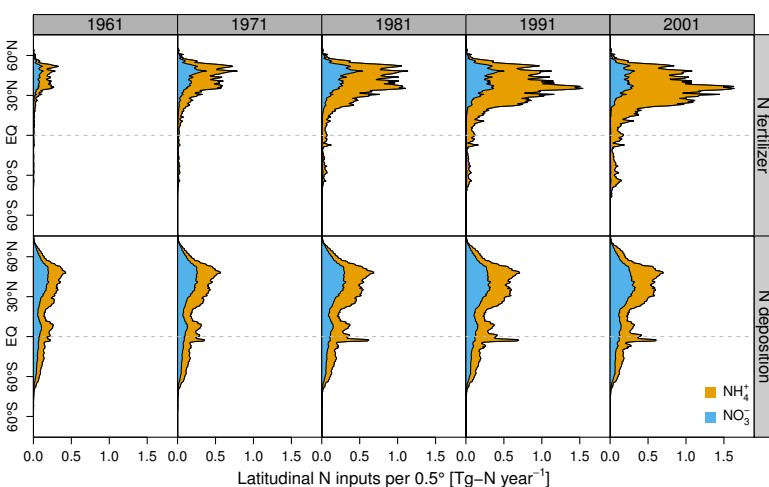

**Figure 11.** Latitudinal $NH_4^+$ and $NO_3^-$ inputs from N fertilizer (upper) and N deposition (lower). Values represent the sum of N inputs at each $0.5°$ resolution. N deposition is from ACCMIP study (Shindell et al., 2013)