# Peer review of "Reconstruction of spatially detailed global map of NH$_4^+$ and NO$_3^-$ application in synthetic nitrogen fertilizer"

_Earth System Science Data, 2016_

## Referee Comment (RC1) · Anonymous Referee #1 · 27 Aug 2016

This manuscript provides a good dataset for nitrogen biogeochemistry study. It made a great effort to impute missing data, and it provides NH4/NO3- ratio, which is unique. I have the following suggestions for the author to revise. 1. The authors used a crop calendar map for the seasonal distribution of fertilizer in a grid, which is good. However, even within a grid, the calendar of the major crop may vary and the fertilizer input vary accordingly. Therefore I suggest author allow crop calendar to vary, for example, a period of 10-15 days. This can be done with a normal distribution, not in a uniform distribution. 2. In the results section, there needs some description on the seasonality of fertilizer input, say, give results of some sample regions. Otherwise it does not match the methodology section. 3. For the second dose of fertilizer, it was not consistent in

the text. Somewhere says 45 days after the first, somewhere says 30 days after the first. 4. Section 3.4 seems not meaningful and thus not necessary. 5. For large countries like US, China and India, fertilizer rate for the same crop may have large regional difference. If possible, for these three large and major nitrogen consumption countries, it is better to obtain N consumption data at subnational level.

---

## Referee Comment (RC2) · Anonymous Referee #2 · 17 Sep 2016

The comment was uploaded in the form of a supplement:
http://www.earth-syst-sci-data-discuss.net/essd-2016-24/essd-2016-24-RC2-supplement.pdf

———————————————

---

## Editor Comment (EC1) · D. J. Carlson (Editor) · 17 Sep 2016

Notes on ESSD 2016-24 and ESSD-2016-35.

Both products achieve the same global resolution (0.5 x 0.5 degrees over global land areas) for approximately similar time periods (1961 to 2010 in one case, 1961 to 2013 in the other case). Both report total synthetic N applied as chemical fertiliser. One elaborates NO3 and NH4 components of the total N, the other adds total P. One starts from country self-reported fertiliser use statistics (from FAO) while the other starts from industry reported fertiliser consumption records (IFA). Both use identical third party crop area data (e.g. Monfreda) but different historical land use data. Both adopt the year 2000 for intercomparison and validation purposes. Both report very similar increases

in total global use of N fertilisers over the time period but they differ slightly in their discussion of geographic and country-specific use patterns over time.

If, as I suspect, both data sets achieve positive reviews, e.g. seem likely to prove useful to readers and subsequent users, and presuming that from the separate review processes ESSD would not designate one or the other data set as preferred, then subsequent users will necessarily need to make a choice between somewhat similar data sets. In that case its seems fair and useful, and a proper use of the open discussion process, to pose a short series of questions to both sets of authors, and to expect that the separate responses should provide a guide to unique aspects and strengths of each data set.

How does the choice of different starting sources, FAOSTAT vs IFA, influence the subsequent processing and overall quality of the derived product?

Does the difference in tactics adopted to deal with variable completeness of country data (imputation to fill gaps in one case and focus on primarily the largest fertiliser users in the other case) induce a substantial or insubstantial difference in the outcomes of the two data production efforts.

Both sets of authors compare their products to Potter et al. 2010 and specifically for the year 2000. If each set of authors now includes the other data set in that comparison, do their overall conclusions change?

What specific information about time histories or geographic patterns of fertiliser use do readers and users gain from the inclusion of NH4 and NO3 data in the one case and from the inclusion of P data in the other case?

Finally, how does each set of authors see their efforts and products as complimentary to the other effort?

---

## Author Comment (AC1) · 19 Sep 2016

Dear anonymous reviewer

Thank you for sharing time to review our manuscript.

Before the reply to your review, I'd like to inform about opening our dataset for all visitors of doi:10.1594/PANGAEA.861203. This dataset is described as $\beta$ version due to the under review.

I'm sorry for this inconvenience.

Kazuya NISHINA

---

## Short Comment (SC1) · 14 Nov 2016

This work aims at generating time-series gridded map of nitrogen fertilizer use rate, application timing and fraction of NH4 and NO3, based on FAO national survey, crop calendar data, crop distribution map, and land use history. It is very important for us to understand the spatial patterns of agricultural chemical fertilizer uses over the past half century. However, I have a few questions regarding its data development:

1) How do you decide whether a region is single or double-cropping based on crop calendar data (Sacks et al., 2010)? As I know, it's impossible for the US to have such large areas of double-cropping agricultural land (over half of country land shown in your figure 1). I am attaching a USDA report on the recent trend of double-cropping in the

US. There are only small percentage of cropland using such practice (about 2% of total cropland in the US). If that's the case, it will substantially affect the cropland areas used in your study and the estimated fertilizer use rate. If harvested area is overestimated or nearly doubled, the fertilizer use rate would be underestimated.

2) You assume there are two fertilizer timing: the first is 7 days before sowing/transplanting and the second is 30 days after based fertilizer application, and the ratio of fertilizer use between these two are 7:3. Is there any supportive evidence or citation for such assumption (timing and ratio to split fertilizer)? In addition, the SAGE dataset has just one average date for each crop in a region. So the dates of N fertilizer inputs were fixed during 1960-2010. Considering the changes in crop distribution and agricultural practices, this timing should vary a lot. Could you discuss what potential consequence would yield by using the fixed application timing?

3) I think the fraction of NH4 in North America may be largely underestimated because the Anhydrous Ammonia and Aqua Ammonia, two important NH4 fertilizer sources in North America, were not included in FAO dataset, but present in USDA data (see table 4 in a single worksheet: http://www.ers.usda.gov/data-products/fertilizer-use-and-price/ and http://ageconomists.com/2016/02/15/nitrogen-fertilizers-shift-happens/) Do you think if it can partially explain why NH4 fraction in North America shown in your figure 6 and 7 is the lowest across the world? I don't know if the same condition also exists in other countries or regions.

Please also note the supplement to this comment:
http://www.earth-syst-sci-data-discuss.net/essd-2016-24/essd-2016-24-SC1-supplement.pdf

---

## Author Comment (AC2) · 16 Jan 2017

**Response to shot comment from Dr. Lu**

NISHINA et al.

January 16, 2017

Dear Dr. C. Lu

Thank you for sharing your time to review our manuscript. We'd like to respond the individual comment one by one.

> 1) How do you decide whether a region is single or double-cropping based on crop calendar data (Sacks et al., 2010)? As I know, its impossible for the US to have such large areas of double-cropping agricultural land (over half of country land shown in your figure 1). I am attaching a USDA report on the recent trend of double-cropping in the US. There are only small percentage of cropland using such practice (about 2% of total cropland in the US). If thats the case, it will substantially affect the cropland areas used in your study and the estimated fertilizer use rate. If harvested area is overestimated or nearly doubled, the fertilizer use rate would be underestimated.

Thank you for your information. Our assumption was just doubling the crop area where the crops in SAGE calendar have information about 2nd or 2 seasons transplanting date. By your comment, we have recognized that our settings of double cropping region were extremely overestimated, globally. As you mentioned, fertilizer rate was underestimated in US (and some other countries) because of our assumption (eq. 4).

So, we have changed the double crop map in the revised version. In the revised N fertilizer map, we utilized the double cropping region based on cropland use intensity (CUI) developed by Siebert et al. (2010, in Remote Sens.). In this map, we used CUI more than 1.3 to be double cropping regions. According to this map, the range of CUI in US varied from 0.9–1.1. Therefore, we avoid the underestimation of N fertilizer rate in US.

As soon as possible, we will upload the revised map in PANGAEA.

[Figure]

[Figure]

Figure 1: Original map

Figure 2: Revised map (Siebert et al., 2010)

2) You assume there are two fertilizer timing: the first is 7 days before sow- ing/transplanting and the second is 30 days after based fertilizer application, and the ratio of fertilizer use between these two are 7:3. Is there any supportive evidence or citation for such assumption (timing and ratio to split fertilizer)? In addition, the SAGE dataset has just one average date for each crop in a region. So the dates of N fertilizer inputs were fixed during 1960-2010. Considering the changes in crop distribution and agricultural practices, this timing should vary a lot. Could you discuss what potential consequence would yield by using the fixed application timing?

Thank you for your important comment. We confirmed the some famers in US to apply N fertilzer at fall especially in dry corn belt region with irrigated area to save their working time. And, also, we confirmed that the timing could affect reactive N gases emission (e.g., Burzaco et al., 2013 in ERL). But, Autumn fertilizer application is not best application practice for all regions even in US and not common in globally. This application methods is only applicable in dry climate region, because of N leaching in follow season. And only specific fertilizer species is recommended in autumn species (http://plantsci.missouri.edu/nutrientmanagement/nitrogen/practices.htm). In addition, we cannot reach the statistics how large proportion of US farmers do this. So, in this time, we don't consider the autumn application in the revised map.

For N fertilizer application, If I have to bring up one, the assumption (criteria) is referenced on the common Maize practice, however, there are no concrete reference.

However, in Robertson and Vitousek (2009, in Annual Review of Environment and Resources), we can see following recommendation;

> Commonly, best practice calls for two applications to field crops, such as corn, with a starter rate (30 kg N ha-1, for example) applied at planting and a side-dress rate (the remaining N to be applied) several weeks later, once the crop has germinated and entered a rapid growth phase.

This is a one of the reference to our assumption. Also, we can referred "Plant nutrition for food security -A guide for integrated nutrient management-" as general guideline for agricultural practice. (ftp://ftp.fao.org/docrep/fao/009/a0443e/a0443e.pdf)

We agreed that our assumption was not realistic to apply the various crop (and vegetable) species management. But, of course, there are no silver bullet. At least, We should excuse such backgrounds in the manuscript.

> 3) I think the fraction of NH4 in North America may be largely underestimated because the Anhydrous Ammonia and Aqua Ammonia, two important NH4 fertilizer sources in North America, were not included in FAO dataset, but present in USDA data (see table 4 in a single worksheet: http://www.ers.usda.gov/data-products/fertilizer-use-and-price/) and
>
> (http://ageconomists.com/2016/02/15/nitrogen-fertilizers-shift-happens/). Do you think if it can partially explain why NH4 fraction in North America shown in your figure 6 and 7 is the lowest across the world? I dont know if the same condition also exists in other countries or regions.

We assumed that other N include $NH_3$ in FAOSTAT. So, the fraction of other N was set to be 100% in $NH_4^+$ and this is not negligible fraction in the consumption (see table 1).

> 3) in your manuscript text, the second application is 30 days after the first application, but in figure 1, it is 45 days after.

Thanks. This is typo. "45 days" is correct. I'd like to revise it in the revision.

**Reference**

Siebert, S., Portmann, F. T., & Döll, P. (2010). Global patterns of cropland use intensity. Remote Sensing, 2(7), 1625-1643.

Burzaco, J. P., Smith, D. R., & Vyn, T. J. (2013). Nitrous oxide emissions in Midwest US maize production vary widely with band-injected N fertilizer rates, timing and nitrapyrin presence. Environmental Research Letters, 8(3), 035031.

Robertson, G. P., & Vitousek, P. M. (2009). Nitrogen in agriculture: balancing the cost of an essential resource. Annual Review of Environment and Resources, 34, 97-125.

---

## Author Comment (AC4) · 16 Jan 2017

**Response to reviewer 1**

**NISHINA et al.**

**January 12, 2017**

Dear anonymous reviewer 1

Thank you for sharing your time to review our manusctipt. We'd like to respond the individual comment one by one as follow;

> 1. The authors used a crop calendar map for the seasonal distribution of fertilizer in a grid, which is good. However, even within a grid, the calendar of the major crop may vary and the fertilizer input vary accordingly. Therefore I suggest author allow crop calendar to vary, for example, a period of 10-15 days. This can be done with a normal distribution, not in a uniform distribution.

We agreed the schedule is better to be fluctuated in some extent. So, we made the new dataset for the fertilizer input date map, which has normal distribution error (as $\sigma = 2$) to the original date.

> 2. In the results section, there needs some description on the seasonality of fertilizer input, say, give results of some sample regions. Otherwise it does not match the methodology section.

Thank you for your suggestion. According to your and reviewer 2's comments, We added the discussion about seasonal changes in N fertilizer input in our dataset.

However, we added latitudinal seasonal N fertilizer input (for both $NH_4^+$ and $NO_3^-$) instead of examples for time-series of regional fertilizer input. This is because the figure for regional time-series inputs in our dataset could just show the spikes (2 or 4 (for double cropping) per year) in each region and this is not so impressive. So, we added the latitudinal time-series data according to the way of display for global time-series data in atmospheric chemistry studies as follow.

[Figure]

Figure 1: Fraction of $NH_4^+$ in N fertilizer input map during 1961–2010

3. For the second dose of fertilizer, it was not consistent in the text. Somewhere says 45 days after the first, somewhere says 30 days after the first.

Thank you for your comments. This is typo. "45 days" is correct. I'd like to revise it in the revision.

4. Section 3.4 seems not meaningful and thus not necessary.

We partially agreed with your suggestion and meaning to the comparison of N fertilizer with N deposition. There are some spatial and temporal inconsistency between them (e.g., fertilizer input only in crop land area).

On the other hand, to date, there are no quantitative comparative reference for $NH_4^+$ and $NO_3^-$ input, respectively. In the view to global N cycling, our comparison could be good start point to recognize how each N input matters in terrestrial ecosystem. In addition, reviewer 2 valued this comparison in the comment. So, we remained this figure in the revised manuscript.

5. For large countries like US, China and India, fertilizer rate for the same crop may have large regional difference. If possible, for these three large and major nitrogen consumption countries, it is better to obtain N consumption data at subnational level.

We fully agreed on your suggestion. We should harmonize more detail information from various regional studies into our N fertilizer map to improve our N fertilizer map.

For example, for US, more spatially detailed N fertilizer map is available in USGS (Gronberg & Spahr, 2012). And, some regional studies in China and Europe —even though they are snapshot for time-series— can be available as more fine spatial resolution of N fertilizer input. However, it is not easy to harmonize these dataset due to the different time- (e.g., just one year) and spatial-scales (e.g., different boundaries) in their map. So, in this time, we acknowledged this insufficiency in the discussion in revised manuscript.

**Reference**

Gronberg, J.M., and Spahr, N.E., 2012, County-level estimates of nitrogen and phosphorus from commercial fertilizer for the Conterminous United States, 1987–2006: U.S. Geological Survey Scientific Investigations Report 2012-5207, 20 p.

---

## Author Comment (AC5) · 30 Jan 2017

**Response to reviwer**

Kazuya NISHINA

January 30, 2017

Dear anonymous reviewer 2

Thank you for sharing your time to review our manuscript and especially for careful reading. We'd like to respond the individual comment one by one as follows;

> The link to FAOSTAT, which necessarily occurs early and prominently through the manuscript, does not resolve. This represents a major barrier to all readers and users. We need a more reliable and permanent link. Either the authors need to convince FAO to deposit a fixed snap shot of the relevant version under a doi at a reliable site (perhaps too much to ask of any specific users) or the authors should include a version of the FAOSTAT data that they used as a component of the Pangaea resource or deposited under some other data authority. Seriously, it makes no sense to cite this source, and almost the entire paper becomes moot, if a reader has no reliable mechanism to start from the same sources. The FAO link must work reliably now and again two years from now. (As a comparison, the links to FAOSTAT global fertiliser data in ESSD-2016-35, which those authors do not represent as their primary source, do work!)

We agreed on your meaning. In fact, for us, it is difficult to follow the updates in FAOSTAT in detail. But, we cannot find the license to re-distribute FAOSTAT data in public. So, at this time, we don't share the original database in open database. However, we would like to inform that we will share the original data when users contact us. FAOSTAT URL in the citation doesn't work now. But, in the manner of website citation, the date of access and URL should be same in the actual access situation. So, we didn't change the citation in the revised manuscript.

> P3L90-92 This statement implies that FAOSTAT includes time series of farmer data (number of farmers?, number, size and types of farms?) at sub-national resolution to allow the authors to successfully manage the changes in national frameworks. We need more information here, to document what the authors used and how the process worked.

Sorry for this. This is typo. The right word is "former" (not farmer). We didn't use the farmer data in this study. So, we revised this in the revised manuscript.

> P4 Description and application of the Amelia data imputation package. The authors provide a careful and useful description of what they assumed and how they proceeded. This user also found the documentation for the Amelia R package adequate and helpful. One wonders, however, whether the application of the imputation idea generally and the specific Amelia code to geographic and temporal patterns of fertiliser data represents a unique and creative solution or a misuse. One could argue that fertiliser application data in fact represent social data (a deliberate human intervention) and therefore the use of Amelia seems quite appropriate? Do the authors know of any other applications of Amelia to these types of more geophysical data sets?

> P4L112 - What are 'panel' data and why do they fit better with the Amelia assumptions?

The structure of FAO dataset is typical country-years dataset (multi-national statistics with time-series). Amelia is developed for dataset with time units for each of N cross-sectional entities such as countries, where often T < N (Honaker and King, 2010). FAOSTAT seem to be typical case. So, Amelia is appropriate for this type of missing data. In addition, already, Amelia imputation methods have been widely applied not only panel data (e.g., IMF, World Bank, and including FAOSTAT) but also in natural science (as cited in the manuscript) in the previous studies. We thought that the most important thing to the validity to use Amelia in this study is Amelia methods being based on the multi-variate distribution, which enable us to think free from the cause-effect relationship among the covariate. So, we explained the basic concept in detail in the manuscript.

Panel data is a technical term in statistics and econometrics. The term panel data (or longitudinal data) refers to multi-dimensional data frequently involving measurements over time.

P4L115 - 'EM' I suppose this acronym refers to expectation-maximization (as in line 98, same page) but the authors should have defined it there or here?

Thanks. We added the complete expression for EM in the revised manuscript.

P4L119 - Do each of these covariants also come from FAOSTAT?

We added the citation for each item.

P4L122 - "in each was" in each what? Nation?

We added "nation" in this sentence. Thanks.

P4L124 - dividing the fertiliser consumption data for each nation for each year by the maximum fertiliser consumption value for that country for the entire period? (Also, I understand why the authors did this scaling but I believe this manipulation to give only lower than existing values from the imputation process deserves mention as part of the processing uncertainties later.)

Yes, we did so. As you pointed out, this procedure limited the upper by the existing values in the national fertilizer consumption. This procedure is on the grounds that the missing values are likely to be found in the early period (during 1961-2010), whereas the statistic in recent year are usually filled in almost all countries. Actually, N fertilizer consumption increase with time. So, this procedure could not cause inappropriate underestimation in N fertilizer consumption in each country.

P5 Section 2.3 - Here the authors provide a useful description of their process for downscaling and for dealing with double cropping. The final statement of the Section, e.g P5L149 seems vague. Do they mean that they applied the double crop weighting for each appropriate crop region for the entire 1961 to 2010 time period or for each time period of double cropping as specified in the SAGE data base?

Thank you for your suggestion. We revised as follow;

The double cropping regions were determined from global crop use intensity (CUI) map developed by Siebert et al. (2010). We defined the double cropping regions in our map as those where the CUI was greater than 1.3. In this region, we doubled the crop area (i.e., the weighting $W_{ij}$ were set to be 2 in the double cropping region) for the entire 1961 to 2010 period.

P5L152 - millet oats should be millet, oats?

**We revised it. Thank you.**

P6L162 - second fertiliser application set to 30 days after initial. However, in Figure 1 on page 16, the authors clearly say 45 days after first fertiliser. The difference of 15 days probably does not make an impact on annual fertiliser usage but the authors should clarify?

**Thank you for your comments.**
**This is typo. "45 days" is correct. I'd like to revise it in the revision.**

P6L171 - "one digit inflation for just one year" Does these mean what we might otherwise call 'order of magnitude', e.g. plus or minus a factor of 10?

**Thanks. We replaced "order of magnitude" instead of "one digit".**

P7L194 - I do not know what the authors mean by "peaky" in this context. Can they give a more precise description?

**We used "discontinuous change" in the revised manuscript.**

P7 Section 3.1 - Here the authors should provide readers and potential users with a more thorough assessment of strengths and weaknesses of imputation approach and of use of Amelia imputation tool. On the one hand, only 16% of countries had missing data and fertiliser use by those countries accounted for an even smaller fraction of total global use. For these reasons the authors conclude that the outcome of the imputation process seems "reasonable". On the other hand, we know that the authors deliberately constrained the imputation process to only produce univariate outcomes - values lower than existing. And we know, as the authors admit and as Figure 3 clearly demonstrates, that missing data did not occur randomly, either in time or geographically. Under these operational constraints, did the large number of iterations (1000) and the use of independent co-variants (GDP, population) in the imputation process reduce or offset the non-random or univariate biases? We need some assurances, or at least a more quantitative assessment, from the authors. Perhaps they know of other applications of Amelia to real world examples that can help us understand the reliability of the outcomes in this case? Reliable outcomes for "all countries"?

Not "only 16% of countries had missing data". 16% of total dataset (totally N = 9811) is missing in national N fertilizer consumption (actually, 73 countries have missing data in N fertilizer during 1961–2010). From your comment, we recognized the sentence, "The countries reported with missing data accounted for a small fraction of N fertilizer consumption compared with global total fertilizer use.", mislead reader to underestimate the importance of missing data. Actually, 5 Tg-N in 2000 of global total N fertilizer consumption was from missing data in our dataset. And this is not the reason to assure the quality of imputation. So we removed this sentence in the revised manuscript.

In the revised manuscript, we explained the effect of transformation in the revised manuscript as follow;

> In the imputation procedure, we used logistic transformation after the scaling N fertilizer consumptions by the observed maximum value of each country for the period 1961–2010. This procedure has the potential to underestimate the total N fertilizer consumption in each country; however, the missing values are likely to occur in the early period (during 1961-2010), whereas statistics for recent years are usually available for almost all countries. In addition, the N fertilizer consumption increases with time. Thus, this procedure can avoid the imputed values being an unreasonable underestimation.

As in the introduction, Amelia had good track records for the imputation of missing data for panel data in various study fields (We added more citation for application of Amelia in the introduction of the revised manuscript). At least, using Amelia, we imputed missing data in an objective, straightforward manner. They are also sufficient excuses to use Amelia to impute missing data.

However, we should agree this method is not perfect solution. So, to share the issues in our application, we added some explanation and discussions for the failure to impute missing data in our study, as follow;

> However, in some countries (e.g., Somalia and Uzbekistan in Fig. 3), the imputed data did not smoothly follow the time-series of national N fertilizer consumption. This was partially because of the abrupt changes in the observed values of those time-series. Changes unrelated to the covariants (i.e., population, crop area) might reduce the accuracy of imputation of missing data. In

fact, we occasionally observed some artifacts even in the reported values (e.g., the sequence of equal values) for the developing countries. In the process of accounting and compiling various datasets, there are inherent uncertainties in the national statistics (Leip, 2010; Winiwarter and Muik, 2010). Such uncertainties could also affect the quality of imputed data; hence, it should be noted that our dataset includes this inevitable uncertainty.

P7,8 Section 3.2. The authors provide a careful, detailed and very useful comparison with Potter 2010 based on year 2000. I note that authors for ESSD-2016-35 made a very similar comparison. This section would represent a very good place for the authors of ESSD-2016-24 to compare their outcomes to ESSD-2016-35? On first glance, the total numbers for N use seem very comparable? (As mentioned earlier, I believe ESSD editors should ensure that this request to authors of ESSD-2016-24 should apply equally to authors of ESSD-2016-35.)

Thank you for your recommendation. We add the comparison with ESSD-2016-35 in the revised manuscript.

P8, Section 3.3. How does or should the inclusion of these $NO_3$ and $NH_4$ data improve our understanding of temporal and geographic patterns of N fertiliser use. The authors of ESSD-2016-24 could make a few clear statements of the value of $NO_3$ and $NH_4$ data compared to total N approach in ESSD-2016-35?

This is our next research topic. The purpose and significance of the inclusion of the ratio in the N fertilizer input are described in introduction.
We prepared the map of the ratio of $NH_4^+$ and $NO_3^-$ in national N fertilizer input in PANGAEA. This is utilized also in the other N fertilizer map. So, we thought that there are small significance in the mention only of ESSD-2016-35.

P8L261 to P9L264. The textual description of temporal changes in $NO_3$ and $NH_4$ use seem substantially in contradiction to Figure 6, and Figure 6 seems inconsistent with Figure 10. From the text here and Figure 10 we learn that "the total amount and fraction of $NH_4$ increased consistently". But looking at Figure 6, and particularly at the global average portrayed in

Figure 6, we must conclude that the ratio of NH4 input to total N input has stayed above 0.8 for the entire 1961 to 2010 period and with only a very narrow variation across those decades? Has this reviewer interpreted the text or the figures incorrectly? Do we in fact have a contradiction inherent in the data as presented?

Thank you for this comment. We mistook the evaluation of global value in Fig. 6. We have just took average in the ratio among regions (shown in Fig. 6). So, this is not actual values in global. The values in Fig. 10 is correct. We removed this incorrect global value from Fig. 6 and replaced the correct ones (see below) in the revised manuscript.

[Figure]

Fig.A Revised figure for the global average

P9L269 - Technically, Figure 11 does not show these total numbers, unless one can integrate the latitudinal data by eye. The authors might consider adding the totals to each panel of Figure 11. In both Figure 10 and Figure 11 the authors should make clear that the values represent the cumulative sum of N inputs while the colours indicate the proportions of NH4 and NO3. I find this overall section quite interesting but the authors might add a sentence or two about the implications of this comparison, to better explain to users why they the users should take an interest in these two sources of reactive nitrogen?

Thank you for your positive suggestion. We added the global total $NH_4^+$ and $NO_3^-$ inputs in both N fertilizer and N deposition for each plot in the revised figure 11. For the $NH_4^+$ and $NO_3^-$, we have already shown colored legends in both figures, however, we added the text explanation in the captions in the revised manuscript.

In Section 3.4, we added a following sentence in the first paragraph to explain the purpose of this comparison.

> N fertilizer and N deposition are the most important sources of disturbance of terrestrial N cycling (Gruber & Galloway, 2008). Although both inputs (finally) comprised of $NH_4^+$ and $NO_3^-$, there has as yet been no quantitative comparison of these two inputs at global.

P9L290 - I think the authors mean that few other sources exist for these kinds of global N data sets. Here the authors might mention ESSD-2016-35 as a comparison?

Thanks. We referred ESSD-2016-35 in this part in the revised manuscript.

P9L292 - Yes, these N data come from national consumption data mapped to crop area, but the process as the authors have described seems quite far from simple.

Indeed, the procedure in developing our map seem to be complex, but N fertilizer rates in each country were determined by simple equation (e.q. 3). For example, our map doesn't consider crop species in N fertilizer rates. So, we wrote "simple" in this sentence.

P10L298 - I think the authors have understated the utility of the NH4 and NO3 data. Those data provide much additional information about national sources and about the time history of use of various forms of N fertiliser.

Thanks.

P10L300 - Yes, the nationally-provided data have uncertainties, but can the authors provide a quantitative estimate to that uncertainty? How do those uncertainties affect the total cumulative use data, e.g. 110 Tg N as in Figure 10. Plus or minus 10%? 20%? In all the figures that follow, only

Figure 3 displays error bars and then only because the imputation run over such a large number of iterations provides statistical uncertainty information. Either here or in the prior discussion the authors should provide their best estimates of cumulative uncertainty from all sources: original data, the imputation process, the crop area estimate, etc. These uncertainty estimates would prove very helpful for users, especially modellers. The statement here about inverse modelling would absolutely require such uncertainty estimates? I find this overall conclusion weak compared to the large effort the authors have put in to assembling and describing this data set. The authors should emphasise the utility of these data for inverse modelling studies, and perhaps compare the strengths (times series, 0.5 degree resolution) and uncertainties in these numbers to the uncertainties around atmospheric N concentrations. There remains a long and somewhat hidden gap between these N input data and the N2O emissions data used in Thompson et al. Also, Winiwarter, cited earlier in the section, addresses uncertainties in nationally reported greenhouse gas inventories. How do those uncertainties apply or compare here? We need a better summary of uncertainties from all sources.

We absolutely agreed on the importance of your comments, here. How the uncertainty matters in global N cycling is really important issue.

However, we cannot provide the consistent uncertainty framework (and values) for each country and global total consumption from this study, because we just evaluated the uncertainty ranges of imputed values in the missing data. As shown in our dataset, the location of missing data in time-series are highly depending on countries. So, our estimation might understate the uncertainties in N fertilizer use at both global and regional scales. In addition, I did not save the imputed dataset (N = 1000) for the summation (The 95% intervals in the imputed data are not entirely symmetric. We need Monte-Carlo integration to sum up them.).

Also, we'd like to emphasize that the uncertainty issue is beyond the scope of this paper (though this manuscript is not research paper). We need further research to address this issue.

P16 Figure 1 - The authors should provide explicit reference to the sources of their crop area data, doubling cropping data, etc. Or link more explicitly to the text where they provide those descriptions? This figure needs better documentation.

Thanks. To clarify these resources, we added the references for these data in this caption.

> P17 Figure 2 - For the US, fertiliser application rates rise but total crop area stabilises or declines so total N fertiliser consumption falls slightly - that make sense. But for China, during a time of expanded crop area and increased fertiliser application rates the total fertiliser consumption appears to fall behind, at least for some years? Have I interpreted this plot correctly? Do the authors have an explanation? Mention the similarity to or emphasise the contrast with Figure 3 in ESSD-2016-35?

Thanks for your comment. This lines are illustrated as a relative units, which calculated by the scaled values in each national N fertilizer consumption and national crop area. So, the crossing the lines in this figure does not mean much in this context. I added the explanation in the caption as follow;

> For both national crop area and N fertilizer consumption, the values are divided by the maximum for this period. So, the unit of fertilizer rates is non-dimension in this figure.

This figure is a just example of procedure and removed all of units for the variables here. For the global average data in ESSD-2016-35, there seem to be no meaningful comparison as far as we thought.

> P18 Figure 3 - Very interesting plot. I believe it conveys a sense of the importance of the covariants because in several cases shown the imputation values clearly do NOT fit the local country time series. The 95% confidence intervals seem quite large in all cases even though of course, by design, they can not exceed values of 1.0; in most cases those 95% CI cover essentially the full range of relative N consumption. Instead of, or in addition to, these specific country examples, could the authors provide a summary of the average error for the 16% of imputation-filled data? This information should help inform the larger uncertainty discussion suggested above?

I'm sorry. We could not understand "summary of the average error for the 16% of imputation-filled data?". Could you clarify this?

> P19 Figure 4 - Not very impressive as log-log plots go. The correlation coefficient and RMSE numbers look good and provide sufficient information. Do we need this plot if we have those numbers in the text or in a small table?

We agreed on your comment. We removed this plot in the revised manuscript.

> P20 Figure 5 - All these numbers presented as absolute, with no uncertainties in either these data or those of Potter 2010?

We clarified the final values for fertilizer map to be obtained the average of the ensemble imputed data in the manuscript. Also, we added the more explanations (how to get, etc) in the caption of this figure.

> P21 Figure 6 - I mentioned already the apparent discontinuity between this data and those presented in the text and in Figure 10. The figure includes some smoothing for each regional data time series, but not explained? Why do so many open circles occur, and why so far above the average, for the North American data which one supposes has reasonably accurate reporting? Explain the boxes: mean plus SE or SD plus max min or quartiles or ...?

Thanks, we revised the discontinuity as explained above.
We added the full explanation for the boxplots in detail.

> P22 Figure 7 - The authors could add a global average number to each panel that should correlate with global data in Figure 6? This figure suggests the small average changes of Figure 6, not the dramatic changes of Figure 10? Pakistan, designated as a hot spot of N use in ESSD-2016-35, does not show up here as particularly important in terms of fraction of NH4 use?

Thanks. We added the global averages in the revised figure.

> P23 Figure 8 - Would we expect to see an offset but repeat pulse in double cropped areas (e.g. of North America or Eurasia) or are the data too smoothed or the application dates too varied? The caption should read "Values represent average NH4 N applied over all crops across each grid cell"?

> P24 Figure 9 - Same question as above about repeat pulses of N inputs observed in areas of double crops, perhaps evident here in April / May (first pulse) and August / September (second pulse) for areas of North America and Eurasia? The caption should read "Values represent average NO3 N applied over all crops across each grid cell"?

Thank you for your comments. As refereed in above response, we clarified values are average for the final product in the material & methods section of the revised manuscript.

Regarding this issue, We revised double cropping region according to Dr. Lu's comment. In addition, according to Reviewer 1 comment, we add new figure (Fig. 9 in the revised ma manuscript) to see seasonal variation of N fertilizer application. Then, we added the discussion for seasonal variation as follow;

> Because of the differences in the $NH_4^+/NO_3^-$ ratio among the countries, there are spatio-seasonal differences in the $NH_4^+$ and $NO_3^-$ inputs throughout the course of a year (Figs. 7–9). During February–March, both inputs exhibit a peak in the northern-hemisphere, especially between 30°N and 60°N (Fig. 9). For the tropics (from the equator to 30°N), both inputs are observed throughout the year. In contrast, in the southern hemisphere, $NH_4^+$ inputs are dominantly observed during September and October in Fig. 9. This is because $NO_3^-$ inputs in the southern-hemisphere countries are a small fraction of the total N fertilizer inputs (Fig. 8). For the double cropping regions (Fig. 1), there are second peaks in both the $NH_4^+$ and $NO_3^-$ inputs around 30°N (Fig. 9), particularly in south China and India (Figs. 7 and 8).

P25 Figure 10 - Make clear that this represents cumulative (NO3 plus NH4) total N with fractions of NO3 and NH4 shown by colours. Explain the differences, if any, between data shown here and data shown in Figure 6.

We fixed it in the revised manuscript.

P26 Figure 11 - Make clear that this represents cumulative (NO3 plus NH4) total N with fractions of NO3 and NH4 shown by colours. Authors could add a cumulative number to each panel to give readers a sense of the integrated totals?

We added global total $NH_4^+$ and $NO_3^-$ inputs in each year in the revised figure.

**Reference**

Leip, A. (2010). Quantitative quality assessment of the greenhouse gas inventory for agriculture in Europe. In Greenhouse Gas Inventories (pp. 245-261). Springer Netherlands.

---

## Author Comment (AC6) · 30 Jan 2017

Dear editor Dr. Carlson

Thank you for your positive response to our manuscript (and dataset) and handing this. It is exciting (but tough for me) to make mutual open discussion between two manuscripts. We would like to response your question and comments.

How does the choice of different starting sources, FAOSTAT vs IFA, influence the subsequent processing and overall quality of the derived product?

[Figure]

The most important feature of this study is to estimate historical $NH_4^+$ and $NO_3^-$ inputs. FAOSTAT has more detail information for n fertilizer species. So, we used FAOSTAT dataset.

From the view to the differences in the amount of N fertilizer inputs between two N fertilizer maps, we thought the choice of database is not so important. Instead of the choice, the different tactics (as you pointed out) resulted in the major difference between two.

> Does the difference in tactics adopted to deal with variable completeness of country data (imputation to fill gaps in one case and focus on primarily the largest fertiliser users in the other case) induce a substantial or insubstantial difference in the outcomes of the two data production efforts.

> Both sets of authors compare their products to Potter et al. 2010 and specifically for the year 2000. If each set of authors now includes the other data set in that comparison, do their overall conclusions change?

We agreed on the advantages in strategy in Lu and Tian (2016), which estimation of N fertilizer rates was based on crop type. However, there are no information about the historical crop area changes in each crop type. As described in Lu and Tian (2016), this is a potential uncertainty source to historical changes in total N fertilizer use. In addition, during the half century, N fertilizer rates in each crop type might change accompanied with economical situation and agricultural technique in each country.

Also, we thought that national policy affect the national N fertilizer consumption (e.g., subsidies for N fertilizer use) in each country. This is a reason to use national census (FAOSTAT) for the estimation of N fertilizer rates in our study.

By this mean, there are positive and negative sides in the both tactics to make N fertilizer map.

Nevertheless (we did comparison in the revised manuscript), there are good general agreement between two (R = 0.84 in spatial correlation in 2000). This is very interesting.

> What specific information about time histories or geographic patterns of fertiliser use do readers and users gain from the inclusion of NH4 and NO3 data in the one case and from the inclusion of P data in the other case?

We provide the dataset for the fraction for $NH_4^+$ ($NO_3^-$) in each country. This can be utilized also in the other N fertilizer maps.

> Finally, how does each set of authors see their efforts and products as complimentary to the other effort?

Our approaches in two papers are quite different to reconstruct global N fertilizer map from existing dataset. They are almost independent each other. We think this is very important fact if we evaluate the uncertainties resulted from N fertilizer input data in global N cycling studies.